# Rice Endosperm Protein Administration to Juvenile Mice Regulates Gut Microbiota and Suppresses the Development of High-Fat Diet-Induced Obesity and Related Disorders in Adulthood

**DOI:** 10.3390/nu11122919

**Published:** 2019-12-02

**Authors:** Yuki Higuchi, Michihiro Hosojima, Hideyuki Kabasawa, Shoji Kuwahara, Sawako Goto, Koji Toba, Ryohei Kaseda, Takahiro Tanaka, Nobutaka Kitamura, Hayato Takihara, Shujiro Okuda, Masayuki Taniguchi, Hitoshi Arao, Ichiei Narita, Akihiko Saito

**Affiliations:** 1Department of Applied Molecular Medicine, Niigata University Graduate School of Medical and Dental Sciences, 1-757 Asahimachi-dori, Chuo-ku, Niigata 951-8510, Japan; n14b140g@mail.cc.niigata-u.ac.jp (Y.H.); kuwahara.s@shc.usp.ac.jp (S.K.); sawako-g@med.niigata-u.ac.jp (S.G.); djkozytoba52@gmail.com (K.T.); 2Division of Clinical Nephrology and Rheumatology, Niigata University Graduate School of Medical and Dental Sciences, 1-757 Asahimachi-dori, Chuo-ku, Niigata 951-8510, Japan; ryoheik@med.niigata-u.ac.jp (R.K.); naritai@med.niigata-u.ac.jp (I.N.); 3Department of Clinical Nutrition Science, Niigata University Graduate School of Medical and Dental Sciences, 1-757 Asahimachi-dori, Chuo-ku, Niigata 951-8510, Japan; hkabasawa@med.niigata-u.ac.jp; 4Clinical and Translational Research Center, Niigata University Medical and Dental Hospital, 1-754 Asahimachi-dori, Chuo-ku, Niigata 951-8520, Japan; ta-tnk@med.niigata-u.ac.jp (T.T.); nktmr@m12.alpha-net.ne.jp (N.K.); 5Division of Bioinformatics, Niigata University Graduate School of Medical and Dental Sciences, 1-757 Asahimachi-dori, Chuo-ku, Niigata 951-8510, Japan; takihara@med.niigata-u.ac.jp (H.T.); okd@med.niigata-u.ac.jp (S.O.); 6Department of Materials Science and Technology, Niigata University, 2-8050 Ikarashi, Nishi-ku, Niigata 950-2181, Japan; mtanig@eng.niigata-u.ac.jp; 7Rice Research Center, Kameda Seika Co. Ltd., 3-1-1 Kameda-kogyodanchi, Kounan-ku, Niigata 950-0198, Japan; h_arao@sk.kameda.co.jp

**Keywords:** childhood nutrition, gut microbiota, kidney disease, metabolic syndrome, obesity, rice protein

## Abstract

Obesity and related disorders, which are increasing in adults worldwide, are closely linked to childhood diet and are associated with chronic inflammation. Rice endosperm protein (REP) intake during adulthood has been reported to improve lipid metabolism and suppress the progression of diabetic kidney disease in animal models. However, the effects of REP intake during childhood on adulthood health are unclear. Therefore, we used a mouse model to experimentally investigate the preconditioning effects of REP intake during childhood on the development of obesity and related disorders in adulthood. Male C57BL/6J mice were pair-fed a normal-fat diet containing casein or REP during the juvenile period and then a high-fat diet (HFD) containing casein or REP during adulthood. Mice fed REP during the juvenile period showed better body weight, blood pressure, serum lipid profiles, lipopolysaccharide (LPS)-binding protein levels, and glucose tolerance in adulthood than those fed casein during the juvenile period. HFD-induced renal tubulo-glomerular alterations and hepatic microvesicular steatosis were less evident in REP-fed mice than in casein-fed ones. REP intake during the juvenile period improved HFD-induced dysbiosis (i.e., *Escherichia* genus proliferation and reduced gut microbiota diversity), thereby suppressing endotoxin-related chronic inflammation. Indeed, REP-derived peptides showed antibacterial activity against *Escherichia coli*, a major producer of LPS. In conclusion, REP supplementation during the juvenile period may regulate the gut microbiota and thus suppress the development of obesity and related disorders in adulthood in mice.

## 1. Introduction

Obesity is a serious medical problem and has been linked to the development of metabolic syndrome and type 2 diabetes [1]. In addition, patients with metabolic syndrome and type 2 diabetes can develop multiple organ complications, including kidney and liver diseases [2,3]. Obesity-related medical costs are rapidly increasing in the United States and have now reached approximately 20% of US health care expenditures [4].

The development of obesity is closely linked to lifestyle habits. In particular, nutrition during the early years of life strongly contributes to the development of obesity and related complications in adolescence and adulthood [5,6]. For example, cohort studies have shown that childhood dietary patterns, such as the intake amounts of vegetables, fruits, and whole grains, are associated with the risk of developing lifestyle disease-related insulin resistance in adulthood [7,8,9].

Rice is a major staple food, especially in Asia. Moreover, rice is the most consumed source of plant protein in Japan [10]. However, food consumption patterns and eating habits among Japanese people, including children, are currently changing [11]. In Asia, diets are moving away from traditional rice-based meals and are becoming increasingly similar to those of Western countries [12]. In particular, rice consumption per energy intake has decreased in children and the proportion of daily fat to total energy intake has increased [12].

Rice endosperm contains about 6% protein. We have previously reported a purification method [13], the nutritional values [14], and various physiological effects of rice endosperm protein (REP). REP contains globulin and the major proteins glutelin and prolamin. Prolamin is found originally in type I protein bodies, while glutelin and globulin accumulate in type II protein bodies in rice endosperm [15]. Glutelin and prolamin can be separated into several protein components by sodium dodecyl sulfate polyacrylamide gel electrophoresis [16]. Notably, REP improves lipid metabolism [17,18,19] and suppresses the progression of diabetic kidney disease in animal models [20]. However, it has not been reported whether and how childhood REP intake influences later life.

The development of obesity and related disorders is associated with alterations in the gut microbiota in humans and mice [21,22]. Diet is critically involved in the regulation of the gut microbiota [23]. In particular, the consumption of a high-fat diet (HFD) decreases gut microbiota diversity [21]. However, no studies have evaluated the influence of REP on the gut microbiota and the development of obesity and related disorders.

Therefore, in this study, we used an animal model to examine whether and how REP administration during childhood regulates the gut microbiota and the development of HFD-induced obesity and related disorders in adulthood.

## 2. Materials and Methods

### 2.1. Preparation of REP and Experimental Diets

A Japanese rice cultivar, *Koshihikari*, was used for the preparation of REP with a purity of 91% crude protein using an alkaline extraction method [13]. The crude protein content of REP was analyzed by the Kjeldahl method using a nitrogen-to-protein factor of 5.95 [24]. A normal-fat diet (NFD; AIN-93G-based, 16% of total energy from fat and 20% from protein) and an HFD (60% of total energy from fat and 20% from protein) were prepared by Oriental Yeast Co., Ltd. (Tokyo, Japan). As indicated in Appendix A, NFD-C and HFD-C used casein as the protein source, whereas NFD-R and HFD-R used REP as its source. HFD-C-AA and HFD-R-AA represent high-fat–based diets that contain amino acid mixtures with amino acid compositions similar to those of casein and REP, respectively, instead of the proteins themselves (Appendix A).

### 2.2. Animal Experiments

Forty 3-week-old C57BL/6J male mice, weighing 9.0–12.0 g, were purchased from CLEA Japan, Inc. (Tokyo, Japan) and were housed under the following conditions: temperature, 24 °C; 12 h light-dark cycle, lights on 08:00–20:00. Following a 1-week adaptation period, the mice were divided into 4 groups (10 mice per group)—CC, CR, RC, and RR—on the basis of body weight (BW). As shown in Figure 1A, the CC and CR groups were fed NFD-C for 6 weeks, while the RC and RR groups were fed NFD-R for 6 weeks (juvenile period) [25]. Subsequently, the CC and RC groups were fed HFD-C, while the CR and RR groups were fed HFD-R for another 12 weeks (adulthood period) [25]; all of these groups were pair-fed based on the dietary intake of the CC group. BW was measured every week, and food intake was evaluated every other day before feeding. After a 12-week period of HFD feeding, a warming holder set to 37 °C was used to restrain the mice for 5 min so that their blood pressure could be measured using the tail-cuff method with a sphygmomanometer (BP-98E; Softron Co., Ltd., Tokyo, Japan). Then, the mice were placed in metabolic balance cages to collect urine for 24 h. The mice were anesthetized with a mixture of 0.3 mg/kg medetomidine, 4.0 mg/kg midazolam, and 5.0 mg/kg butorphanol administered by intraperitoneal injection, after which blood samples were taken via the inferior vena cava. In addition, the kidney, liver, and epididymal, right inguinal subcutaneous, mesenteric, and retroperitoneal fat were excised and immediately weighed. The right kidneys and livers were morphologically examined. The same feeding protocol for the above animal experiment was separately adopted for oral glucose tolerance tests (OGTTs).

The effects of casein and REP amino acid compositions were examined as follows. Nine-week-old wild-type C57BL/6J male mice (CLEA Japan, Inc., Tokyo, Japan) were allowed to adapt to our facility’s conditions for 1 week, during which they were fed a standard diet. After adaptation, the mice were randomly assigned to body weight-matched groups (*n* = 10, each) and were pair-fed HFD-C-AA or HFD-R-AA for 12 weeks. REP and casein amino acid compositions are shown in Appendix A. Food intake was evaluated every other day before feeding. At the end of the experiment, the mice were anesthetized and the liver and epididymal, right inguinal subcutaneous, mesenteric, and retroperitoneal fat were excised and immediately weighed. 

All animal experiments in this study were conducted in accordance with the National Institutes of Health Guide for the Care and Use of Laboratory Animals and were approved by the Animal Care and Use Committee and the Animal Experiment Ethics Committee of Niigata University (approved on 26 March 2015 and on 21 July 2015; approval numbers: 26-372-1 and 27-133-3, respectively).

### 2.3. Blood and Urine Analysis

Total cholesterol, triglyceride (TG) and creatinine in the serum, and albumin and creatinine in the urine were measured at Oriental Yeast Co., Ltd., using a 7170 Automatic Analyzer (Hitachi, Ltd., Tokyo, Japan). The serum levels of leptin, advanced glycation end-products, and lipopolysaccharide-binding protein (LBP) were measured using enzyme-linked immunosorbent assay (ELISA) kits produced by the Morinaga Institute of Biological Science, Inc. (Tokyo, Japan), Cell Biolabs (San Diego, CA, USA), and HyCult Biotechnology (Uden, The Netherlands), respectively, according to the manufacturers’ instructions.

### 2.4. Cytokine Quantification in Serum and Tissues

Frozen tissues (kidney and liver) were weighed and homogenized in tissue lysis buffer (AimPlex Biosciences, Inc., Pomona, CA, USA). The tissue lysates were incubated on ice for 5 min and centrifuged at 14,000× *g* for 10 min at 4 °C before the supernatant fraction was collected for assays. Serum and tissue lysate levels of tumor necrosis factor-α (TNF-α), interleukin (IL)-1β, IL-6, and monocyte chemoattractant protein-1 (MCP-1) were determined with a panel kit (AimPlex Biosciences, Inc) by flow cytometry analysis using a BD FACSCalibur system (BD Biosciences, San Jose, CA, USA).

### 2.5. Analysis of Kidney Morphology

Histological damage was assessed as follows. The right kidney was fixed in 4% paraformaldehyde phosphate buffer solution (Fujifilm Wako Pure Chemical Corporation, Osaka, Japan) immediately after resection. The kidney was embedded in paraffin, sectioned into 4-μm–thick slices, and stained with periodic acid-Schiff stain. The mesangial matrix and glomerular areas and the number of proximal tubular vacuoles (dysfunctional autolysosomes) were evaluated as described previously [26].

### 2.6. Analysis of Liver Morphology and TG Content

Liver tissues were prepared for paraffin embedding. Tissue slides were then stained with hematoxylin and eosin. The morphology of the liver was evaluated under light microscopy after mounting. Total lipids were extracted from the liver, as previously described [27], and TG content was measured by an enzymatic method using a specific kit (Fujifilm Wako Pure Chemical Corporation).

### 2.7. Metabolic Assessment

Metabolic rate was measured by indirect calorimetry using an open-circuit Oxymax system (Columbus Instruments, Columbus, OH, USA) according to the manufacturer’s instructions. The mice (22 weeks old) were transferred to sealed open-circuit chambers and maintained under the same housing conditions (at 24 °C under a 12 h light-dark cycle; lights on 8:00–20:00) with the same food intake and water available *ad libitum*. The mice were weighed and acclimatized to the chambers for 24 h before having their physiological parameters automatically recorded every hour for 24 h. Room air was passed through the chambers at a flow rate of 0.4 L/min. Exhaust air from each chamber was sampled at 10 min intervals for 90 s. O_2_ and CO_2_ contents, which were used to estimate oxygen consumption (VO_2_) and carbon dioxide production (VCO_2_), were determined by passing the sampled air sequentially through O_2_ and CO_2_ sensors (Columbus Instruments). Metabolic rate (kcal/h) was calculated using the following equation: (3.815 + 1.232 × respiratory exchange ratio) × VO_2_, where the respiratory exchange ratio represents the volume of CO_2_ produced (mL/kg body weight/h) per volume of O_2_ consumed (mL/kg body weight/h), and VO_2_ is the volume of O_2_ consumed per hour.

### 2.8. Determination of Fecal Total Lipid and Bile Acid Levels

Fecal lipids were extracted as described by Folch and Lees [28]. Methyl esterification of fatty acids (palmitic acid, stearic acid, oleic acid, and linoleic acid) was carried out using a Fatty Acid Methylation Kit (Nacalai Tesque, Kyoto, Japan), and fatty acids were analyzed on an Agilent 7890 GC-MS with CP-Sil 88 for FAME column (Agilent, Palo Alto, CA, USA). The initial oven temperature of 100 °C was increased to 240 °C (3 °C/min) and maintained for 15 min. One microliter of sample was injected into the GC/MS system. Fecal total bile acids were also determined using a Total Bile Acids Test Wako Kit (Fujifilm Wako Pure Chemical Corporation) after ethanol extraction.

### 2.9. 16S rRNA Analysis of Fecal Samples and Bioinformatics Analysis

The gut microbiota was investigated by 16S rRNA gene sequencing of fecal samples at 10 and 22 weeks. DNA was isolated using a Nucleo Spin^®^ DNA Stool Kit (Sigma-Aldrich, St. Louis, MO, USA) according to the manufacturer’s instructions. 16S rRNA analysis of fecal samples was performed as described previously [29] at Takara Bio (Shiga, Japan). For bioinformatics analysis, QIIME software (version 1.8.0) was used to merge the forward and reverse reads of each FASTQ file [30]. Taxonomy annotation was performed as described previously [31] based on the taxonomy information registered in RDP [32] (downloaded on 29 August 2017). Relative abundance was calculated by dividing the number of reads assigned to each taxonomic class by the total number of reads in each sample. As a diversity measure, the Shannon index was calculated for each sample as previously described [33]. In addition, we performed principal coordinate analysis (PCoA). Euclidean distance was calculated using the genus relative abundance in each sample. PCoA was calculated with the cmdscale function in the R “stats” library (https://www.R-project.org/). Finally, Spearman’s correlation coefficients were calculated between the genus relative abundance in each sample and clinical information. Hierarchical clustering and heat map analysis were performed with the default setting using the pheatmap library for R (http://CRAN.R-project.org/package=pheatmap).

### 2.10. OGTT

OGTTs were performed as follows: mice fasted for 6 h were administered glucose by gavage (2 g/kg glucose, 20% glucose solution), and blood samples were taken via the tail vein before administration and at 15, 30, 45, 60, 90, and 120 min after. Blood glucose was measured using a Glutest Neo Alpha device (Sanwa Kagaku Kenkyusho Co., Ltd., Aichi, Japan). Plasma insulin was measured using a Mouse Insulin ELISA Kit (Morinaga Institute of Biological Science, Inc.). The homeostatic model assessment of insulin resistance (HOMA-IR) was determined using fasting glucose and insulin plasma levels: glucose (mM) × insulin (pM)/22.5.

### 2.11. Measurement of the Antimicrobial Activity of Peptides Derived from Casein and REP

Hydrolysates of casein and REP were prepared as previously reported [34]. Hydrolysates including casein or REP peptide were separated into 20 fractions by ampholyte-free isoelectric focusing (autofocusing) as previously described [35,36]. *Escherichia coli* ATCC 25922 cells were precultured overnight at 37 °C in LB medium. The antimicrobial activity of the hydrolysate fractions against *E. coli* ATCC 25922 was measured using 96-well plates as previously reported [35,36]. The data are expressed as the mean ± standard deviation of three individual experiments.

### 2.12. Statistical Analysis

Data are expressed as means ± standard deviation. We used a linear mixed-effects model to evaluate the slope of change in BW over time as well as the interaction between time (weeks) and the combinations of juvenile-period and adulthood protein source (casein or REP). In addition, two-way analysis of variance (ANOVA) for unpaired repeated measures was performed to determine the interaction between the effects of protein source during the juvenile period and adulthood as independent variables on parameters measured at 22 weeks of age as dependent variables. This analysis was followed by multiple comparison testing using Tukey’s correction as post-hoc analysis to identify differences. The results of the mice fed an HFD with amino acids were analyzed by Student’s *t*-test. All statistical analyses were performed using GraphPad Prism statistical software version 5 (GraphPad Software, Inc., San Diego, CA, USA) and SPSS statistical software version 22 (IBM Japan, Tokyo, Japan). Statistical significance was set at *p* < 0.05.

### 2.13. Data Availability

The sequences reported in this paper have been deposited in the DDBJ sequence read archive (accession no. DRA007335).

## 3. Results

### 3.1. REP Administration during the Juvenile Period Suppresses the Development of Obesity and Related Disorders in Adulthood

The values of the parameters measured at 22 and 10 weeks of age are shown in Table 1 and Table 2, respectively. There were no significant differences in physical and biochemical parameters at 10 weeks between the mice fed casein or REP (Table 2). As shown in Figure 1B, the slope of BW change from week 4 to week 22 was significantly lower in the groups receiving REP during the juvenile period and/or adulthood. REP intake during HFD feeding in adulthood was significantly associated with decreases in systolic blood pressure, serum TG, total fat and mesenteric fat weight, and BW at 22 weeks compared with casein intake in the same period (Table 1), as previously reported [20]. Of note, REP intake during the juvenile period before starting HFD feeding was also significantly and independently associated with decreases in the above parameters at 22 weeks compared with casein intake. In addition, as shown in Figure 1C and Table 1, REP intake during adulthood was significantly associated with decreases in fasting blood glucose, plasma insulin, and HOMA-IR compared with casein intake during adulthood. Notably, REP intake during the juvenile period was also significantly and independently associated with decreases in the above parameters as well as the area under the curve (AUC) for glucose in the OGTTs. Thus, protein sources during the juvenile period influence the development of HFD-induced obesity and related disorders in adulthood.

### 3.2. Administration of REP during the Juvenile Period Lowers Inflammation Markers in Adulthood

In general, chronic low-grade inflammation is observed in obesity [37]. The effects of REP administration on inflammation markers in the serum, kidney, and liver in HFD-induced obese adult mice are shown in Table 3. As previously reported [18], REP intake during adulthood was significantly associated with a lowering of the serum levels of LBP, leptin, IL-1β, and IL-6, and the kidney levels of IL-6, compared with casein intake during adulthood. However, REP intake during the juvenile period was more significantly and independently associated with a lowering of the serum levels of LBP and the serum, kidney, and liver levels of leptin, IL-1β, IL-6, TNF-α, and MCP-1 compared with casein intake during the juvenile period.

### 3.3. Administration of REP during the Juvenile Period Suppresses the Development of HFD-Induced Kidney and Liver Disease in Adulthood

HFD-fed mice develop kidney disease with increased kidney weight and albuminuria [26]. Light microscopy analysis was performed to investigate HFD-induced glomerular hypertrophy, glomerular mesangial matrix expansion, and vacuolization in the proximal tubules, which are also observed in obesity-related kidney disease in humans [38]. Vacuolization in the proximal tubules is indicative of autolysosomal dysfunction mediated by megalin, an endocytic receptor [26]. As shown in Table 4 and Figure 2A–C, REP intake during adulthood was significantly associated with lower urinary albumin excretion and decreased glomerular area compared with casein intake during adulthood, as previously reported [20]. Furthermore, mesangial matrix expansion and tubular vacuolization area were larger in the CC group than in the CR group. Importantly, REP intake during the juvenile period was significantly and independently associated with lower kidney weight and urinary albumin excretion and a decrease in the glomerular area compared with casein intake during the juvenile period. In addition, mesangial matrix expansion and tubular vacuolization were more evident in the CC group compared with the RC group. No significant differences in creatinine clearance were found among the groups.

HFD-fed mice are reported to show liver hypertrophy and fat accumulation [27]. Similar results were also seen in the present study. Liver weight was higher in the CC group compared with the CR and RC groups (both *p* < 0.001 by two-way ANOVA with post-hoc Tukey’s analysis). REP intake during the juvenile period or adulthood was significantly associated with decreased TG content in the liver compared with casein intake in the same period (Table 4). Histological analysis revealed that REP intake during the juvenile period before the introduction of HFD feeding alleviated HFD-induced microvesicular steatosis of the liver in adulthood (Figure 2D).

### 3.4. No Differences in Energy Consumption among the CC, RC, CR, and RR Groups

To define the mechanisms underlying the difference in the effects of protein source (REP or casein) on HFD-induced weight gain, we evaluated energy intake and consumption by placing 22-week–old mice in metabolic cages for 24 h. No significant differences were found in energy intake (data not shown), VO_2_, respiratory exchange ratio, and energy consumption among all pair-fed groups (Appendix A).

### 3.5. Administration of REP Regulates Lipid and Total Bile Acid Excretion in Feces

High excretion of lipids and fatty acids in the feces of mice fed REP with an HFD has been reported [18]. Total lipid excretion in feces was lower in the CC group compared with the CR and RC groups (both *p* < 0.001 by two-way ANOVA with post-hoc Tukey’s analysis). REP intake during HFD feeding in adulthood was significantly associated with an increase in fatty acids and total bile acid in feces compared with casein intake in the same period, as previously reported [18]. However, REP intake during the juvenile period before HFD initiation was significantly associated with an increase in palmitic acid, total saturated fatty acid, and total bile acids compared with casein intake (Appendix A).

### 3.6. Differences in the Amino Acid Composition of REP and Casein Are not Associated with the Inhibitory Effect of REP on HFD-Induced Obesity

Whether the effect of REP on the suppression of HFD-induced obesity was due to amino acid composition was investigated using mice pair-fed HFD-C-AA or HFD-R-AA with an amino acid mixture composition of casein or REP, respectively. No significant differences were found in BW, total fat weight, and mesenteric fat weight between the two groups (Figure 3A–D). Furthermore, no differences in kidney and liver weight and systolic blood pressure were found between the two groups (Figure 3E–G). These results suggest that REP-derived peptides, but not amino acids, which may be produced by digestion in the gut, are involved in suppression of HFD-induced obesity and related disorders.

### 3.7. Administration of REP during the Juvenile Period Suppresses the Development of HFD-Induced Dysbiosis of the Gut Microbiota in Adulthood

The gut microbiota may strongly influence the development of HFD-induced obesity and related disorders [21]. To characterize the fecal microbiota of mice fed different proteins, we sequenced the variable regions 3–4 (V3–V4) of bacterial 16S rRNA genes present in fecal samples collected at 10 and 22 weeks of age. Phylum-level analysis indicated that the fecal microbiota at 22 weeks of age comprised mainly three phyla—Actinobacteria, Bacteroidetes, and Firmicutes—and that the abundance ratios of these three phyla in the fecal microbiota differed according to the ingested protein source (Figure 4A). Genus-level analysis of the fecal microbiota at 22 and 10 weeks of age is shown in Figure 5A,B. A PCoA plot with unweighted UniFrac distances between all 40 fecal samples showed segregation between protein source during adulthood for phylum (Figure 4B). To investigate how dietary proteins and HFD affected microbiota phylogenetic richness, we analyzed α-diversity, reflected by rarefaction and phylogenetic diversity. As shown in Figure 4C, REP intake during the juvenile period before the introduction of HFD feeding was also significantly associated with increased α-diversity compared with casein intake. The abundance ratio of the *Escherichia* genus was highest in the CC group compared with the other groups (Figure 4D). Spearman’s rank correlations were used to identify genera whose relative abundance at 22 weeks of age was strongly correlated with growth status, blood and urine parameters, and histological findings of the kidney affected by HFD (Figure 5C). As shown in Appendix A, in the genus-level analysis, two-way ANOVA of the abundance ratio of five genera showed a significant effect of juvenile-period protein source. Furthermore, Tukey’s multiple comparison test revealed significant differences in the abundance ratio of the other four genera between the CC and RC groups. In addition, *Escherichia* genus levels were significantly correlated with BW, HOMA-IR, and TNF-α in the kidney and liver, and with urinary albumin excretion, kidney weight, serum LBP, and histological parameters in the kidney. The fecal microbiota were also examined at 10 weeks of age (before HFD initiation). The phylum-level and PCoA plot results showed differences between the two sources of protein (REP or casein) (Figure 4E,F). Furthermore, the fecal microbiota of REP-fed mice showed higher α-diversity and lower *E. coli* levels compared with casein-fed mice (Figure 4G,H).

### 3.8. REP-Derived Peptides Show Antibacterial Activity

As mentioned above, REP-derived peptides may exert biological effects on the gut microbiota. Accordingly, autofocusing was used to separate hydrolyzed REP and casein into 20 fractions, after which *E. coli* was used to determine the antimicrobial activity of each fraction. None of the casein hydrolysate fractions exhibited significant antimicrobial activity against *E. coli* (Figure 6A). In contrast, fractions 1, 19, and 20 of the hydrolyzed REP showed significant antimicrobial activity against *E. coli* (Figure 6A). These fractions also exhibited concentration-dependent antimicrobial activity against *E. coli* (Figure 6B). In addition, the antimicrobial activity of each fraction against *E. coli* was observed in agar diffusion assays (Figure 6C). These results suggest that REP hydrolyzed in the intestines may be able to decrease *E. coli* levels and maintain an appropriate state of the gut microbiota.

## 4. Discussion

This study showed that, compared with casein, the administration of REP to mice during the juvenile period suppresses the development of HFD-induced obesity and related disorders in adulthood. The beneficial effects of REP administration along with an HFD during adulthood were similar to those previously reported [17,18,19,20]. REP administration during the juvenile period is likely to be involved in suppressing the growth of Proteobacteria (*Escherichia*), which would reduce the production of LPS, an endotoxin, and reduce the low-grade inflammation associated with insulin resistance. Interestingly, such a preconditioning effect of REP intake during the juvenile period appears to persist with an HFD during adulthood even after the dietary protein is changed to casein.

Childhood dietary patterns are clinically associated with risk of developing obesity and related diseases in adulthood in humans [39]. A higher quality diet in adolescence along with dietary improvements are related to the prevention of obesity during the transition from adolescence to adulthood [39]. Interestingly, the early life stage has been identified as an important period in which the gut microbiota evolve towards an adult-like state, which is thought to be closely linked to diet during this period [40]. Hence, the novelty of this study is that we identified possible mechanisms by which REP intake during childhood might help to maintain gut microbiota diversity and prevent obesity and related disorders in adulthood.

Using 16S rRNA pyrosequencing, we found that REP intake suppressed the HFD-induced increase in Proteobacteria in the gut (only the *Escherichia* genus was detected among Proteobacteria in our experiments). Increased levels of LPS, which is an endotoxin derived from Gram-negative bacteria, including Proteobacteria, can induce inflammation and is associated with the development of obesity and its complications [41,42]. Moreover, the levels of Proteobacteria (*Escherichia*) were significantly and negatively correlated with BW, insulin resistance, inflammatory cytokine levels, and kidney injury in this study. Recently, the role of the gut microbiota in the development of obesity and associated disorders (e.g., metabolic syndrome and type 2 diabetes) has also been studied [38]. Qin et al. [43] reported that the gut microbiota in patients with type 2 diabetes showed higher levels of Bacteroidetes and *E. coli* compared with controls. In another study, the gut microbiota of obese and type 2 diabetic patients had a high abundance of Proteobacteria [44]. Similar findings linking the gut microbiota to fat mass and BW have been observed in diet-induced obese mice [45]. In addition, when lean mice were switched to an HFD, there was an expected decrease in Bacteroidetes and an increase in Proteobacteria and Firmicutes [45]. Systemic chronic low-grade inflammation also plays a key role in metabolic disorders associated with obesity involving insulin resistance [38]. The levels of LPS and LBP are significantly higher in the serum of obese individuals [46] and patients with type 2 diabetes than in controls [47], respectively. Moreover, many reports have indicated that LPS causes kidney disease [48]. Accordingly, Proteobacteria-derived LPS is associated with systemic chronic low-grade inflammation and an increase in the serum levels of LPS can be indicative of an imbalance in the microbiome in obesity [22]. Furthermore, the gut microbiota map of non-vegetarians has a higher abundance of *E. coli* compared with that of vegetarians [44]. Therefore, we suggest that there is a link between the decrease in abundance of Proteobacteria (*Escherichia*) and the intake of REP or a rice-based diet.

Notably, food-derived proteins or peptides, such as specific whey proteins, are reported to have antimicrobial activity [49]. Digestion also facilitates the formation of potent antimicrobial whey-derived peptides, including the pepsin-catalyzed conversion of lactoferrin to lactoferricin [49]. Taniguchi et al. [35] characterized cationic peptides from enzymatic hydrolysates of REP and determined the antimicrobial activity of each fraction against pathogenic microbes. In the present study, we confirmed that peptide fractions derived from enzymatic hydrolysates of REP have antimicrobial activity against *E. coli* in vitro, suggesting that REP digestion-derived peptides may regulate the diversity of the gut microbiota. We examined the effects of the amino acid compositions of casein and REP, but there were no significant differences in body, fat, liver, and kidney weights in mice (Figure 3). Thus, in this study, we suggest that the effects of REP intake on the alteration of the gut microbiota and the prevention of obesity and related disorders were not due to differences in amino acid composition but were due to the properties of REP digestion-derived peptides.

The beneficial effects of REP have been described, such as improvement in lipid metabolism and prevention of diabetic nephropathy in adulthood animal models [17,19,20]. In this study, REP intake also ameliorated lipid metabolism disorders and kidney disease induced by an HFD during adulthood. Moreover, the excretion of lipids, fatty acids, and bile acids in the feces of mice fed REP, especially during adulthood, was higher than in mice fed a casein-based diet. It has been reported that REP can bind to fatty acids and bile acids and interfere with their absorption in the lower intestine and thereby may have a specific ability to increase their fecal excretion [18], as reported for soybean peptides [50]. The mechanism of action of cholestyramine, a basic anion-exchange resin used to improve lipid metabolism, involves its binding to bile acids and increasing their excretion in feces in humans [51] and in animal models [52]. It is thus suggested that REP or REP digestion-derived peptides may similarly adsorb lipids and bile acids, increase their fecal excretion, and suppress lipotoxicity in vivo [18].

An important limitation of our study is that we used only an animal model. Further studies using different animal models may be needed to verify our findings. However, instead of conducting further animal research, it would be preferable to start human trials with REP. REP has been administered safely to human patients undergoing dialysis [53], and it has also shown beneficial effects on metabolic syndrome in adults [54]. Therefore, we suggest that the intake of REP or rice-based diets in childhood may prevent obesity and related disorders in adult humans. In addition, we are planning additional in vitro studies to identify potential biochemical properties of REP and REP digestion-derived peptides.

## 5. Conclusions

The administration of REP during the juvenile period can suppress the development of HFD-induced obesity and related disorders in adulthood in mice. The present results indicate that the gut microbiota of mice can be altered by different dietary proteins, particularly during the juvenile period. Furthermore, the reduction of the abundance of Proteobacteria (*Escherichia*) in the gut induced by REP administration is likely to suppress endotoxin-related chronic inflammation. In addition, REP or its peptide products may exert lipid-absorption effects in the gut and promote fecal lipid excretion. Further studies in humans are needed to assess the clinical effects of REP and to recommend a rice-based diet during childhood.

## Figures and Tables

**Figure 1 nutrients-11-02919-f001:**
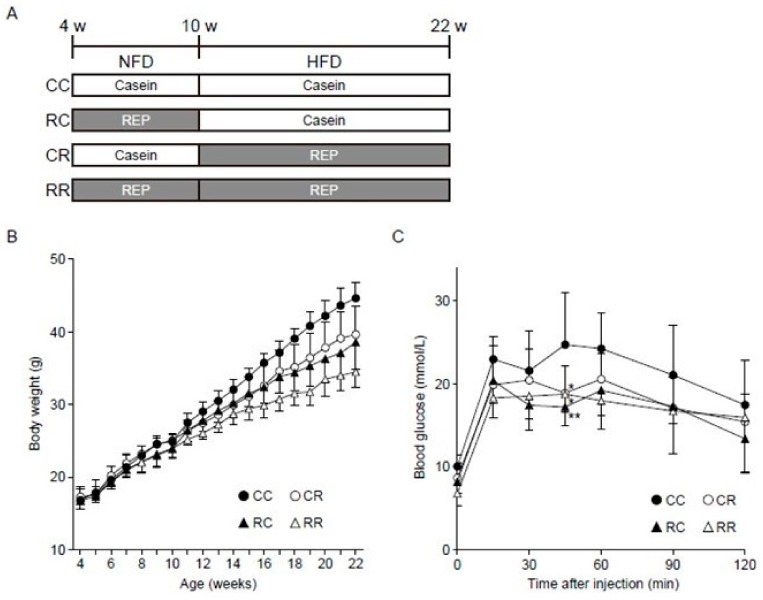
Effects of rice endosperm protein (REP) intake on body weight and oral glucose tolerance in mice. (**A**) Schematic overview of the experimental design, including study groups according to the ingested protein source. Male C57BL/6J mice (four weeks old) were pair-fed a normal-fat diet (NFD) with casein or REP for six weeks (juvenile period) and then a high-fat diet (HFD) with casein or REP for 12 weeks (adulthood period). The mice were divided into four groups (CC, RC, CR, and RR). (**B**) Linear mixed-effects models revealed that the interaction between time (weeks) and the combinations of juvenile-period and adulthood protein source (casein or REP) was significant, showing that the groups receiving REP during the juvenile period had significantly reduced weight gain during adulthood (*p* < 0.001). (**C**) Oral glucose tolerance tests in the mice of the four groups (*n* = 10 per group). Results of the analyses of fasting blood glucose and the area under the curve are shown in Table 1 REP intake during HFD administration in adulthood was significantly associated with a decrease in blood glucose levels at 15 and 60 min compared with casein intake in the same period. Furthermore, blood glucose levels at 45 min were higher in the CC group than in the CR group (*p* = 0.016 by two-way analysis of variance with post-hoc Tukey’s analysis). However, REP intake during the juvenile period before the introduction of HFD feeding was significantly associated with a decrease in blood glucose levels at 30 and 60 min compared with casein intake in the same period. In addition, blood glucose levels at 45 min were higher in the CC group than in the RC group (*p* = 0.001 by two-way ANOVA with post-hoc Tukey’s analysis). All results are shown as the mean ± standard deviation. * *p* < 0.05 and ** *p* < 0.01.

**Figure 2 nutrients-11-02919-f002:**
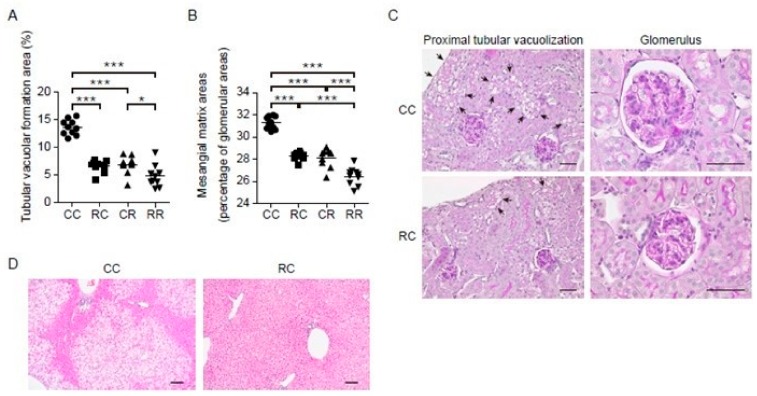
Impact of REP intake during the juvenile period on morphological findings of the kidney and liver in HFD-induced obese mice. Proximal tubular vacuolar formation area (**A**) and mesangial matrix area (**B**) were measured using Image-Pro and were analyzed by two-way analysis of variance (ANOVA) in 10 randomly selected fields of view (*n* = 10 in each group). There were interaction effects of protein source during the juvenile period and adulthood for proximal tubular vacuolar formation area and mesangial matrix area (both *p* < 0.001). Proximal tubular vacuolar formation area and mesangial matrix expansion were larger in the CC group than in the CR group and the RC group (both *p* < 0.001 by two-way ANOVA with post-hoc Tukey’s analysis). (**C**) Representative photomicrographs illustrating glomeruli and proximal tubular vacuolar formation in the cortical regions of the CC and RC groups (periodic acid-Schiff staining). HFD-induced glomerular hypertrophy, glomerular mesangial expansion, and vacuolization in the proximal tubules (arrows) were observed. Vacuolar formation was particularly evident in the CC group, which was significantly increased compared with the RC group. Glomerular hypertrophy was seen in the CC group but was suppressed in the RC group. (**D**) Representative hematoxylin and eosin-stained sections of liver tissues in the CC and RC groups. Fat was removed by paraffin embedding, and lipid droplets are observed as vacuoles. Vacuolar areas were larger in the CC group and smaller in the RC group. Scale bar represents 50 μm. * *p* < 0.05 and *** *p* < 0.001.

**Figure 3 nutrients-11-02919-f003:**
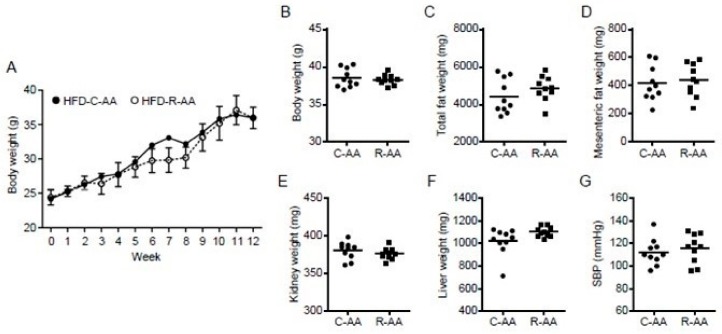
Differences in the amino acid compositions of REP and casein have no effect on body weight and other parameters. (**A**) Body weight was measured every week. (**B**–**G**) Final body weight, systolic blood pressure (SBP), total fat, mesenteric fat weight, liver weight, and kidney weight were measured at the end of the experiment. No differences were found in any parameters between the two groups. Data are shown as means ± standard deviation. C-AA, HFD with an amino acid mixture composition of casein; R-AA, HFD with an amino acid mixture composition of REP.

**Figure 4 nutrients-11-02919-f004:**
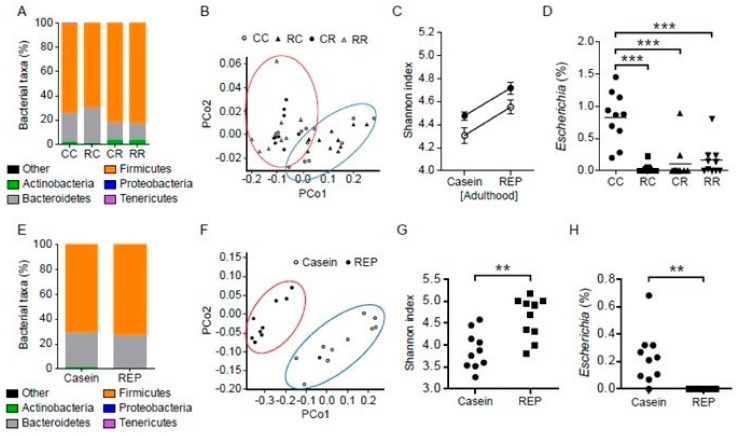
Effects of REP administration during the juvenile period on the composition of the gut microbiota of mice. 16S rRNA gene analysis revealed phylum-level differences in the gut microbiota of mice fed an NFD or HFD with casein or REP. These results show the fecal microbiota community of mice at 22 weeks of age (**A**–**D**). (**A**) Relative abundance in the mouse gut microbiota at the phylum level. (**B**) Principal coordinate analysis (PCoA) of gut microbiota clustering. The variation percentage indicated by the plotted principal coordinates is shown by the axis labels. (**C**) The rarefaction depth used was 60,000 sequences per sample, and the Shannon index was rarefied to 37,810 reads per sample prior to analysis. REP intake during HFD feeding in adulthood was significantly associated with increased α-diversity compared with casein intake in the same period. However, REP intake during the juvenile period before the introduction of HFD feeding was also associated with increased α-diversity compared with casein intake (two-way analysis of variance: effect of juvenile-period protein source, *p* < 0.001; effect of adulthood protein source, *p* = 0.005; interaction, *p* = 0.971). Closed circles, REP during the juvenile period; open circles, casein during the juvenile period. (**D**) There was an interaction effect in the relative abundance of the *Escherichia* genus in the feces of the groups. The abundance ratio of the *Escherichia* genus was higher in the CC group than in the CR and RC groups (both *p* < 0.001 by two-way ANOVA with post-hoc Tukey’s analysis). (**E**–**H**) Relative abundance at the phylum level (**E**), PCoA of clustering (**F**), Shannon index (**G**), and relative abundance of the *Escherichia* genus (**H**) in the gut microbiota of mice at 10 weeks of age. All results are reported as the mean ± standard deviation (*n* = 10 in each group). ** *p* < 0.01, and *** *p* < 0.001.

**Figure 5 nutrients-11-02919-f005:**
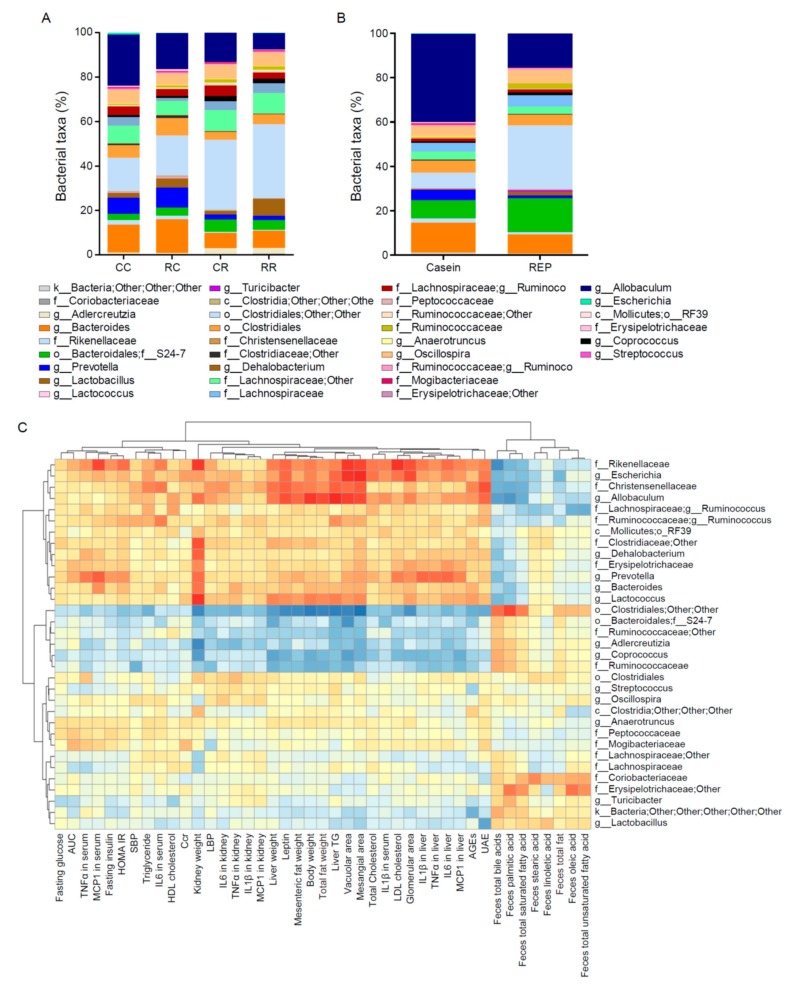
Effect of REP intake on the gut microbiota composition of mice with diet-induced obesity and correlations between microbiota abundance and parameters. (**A**,**B**) Relative abundance in the microbiota of mice at the genus level at 22 weeks of age (**A**) and at 10 weeks of age (**B**). (**C**) Heatmap of Spearman’s correlations between all data and relative genus abundance at 22 weeks of age. SBP, systolic blood pressure; AUC, the area under curve (AUC) for glucose during OGTT; HOMA-IR, homeostatic model assessment of insulin resistance; AGEs, advanced glycation end-products; UAE, urinary albumin excretion; CCr, creatinine clearance; TG, triglyceride; LBP, lipopolysaccharide-binding protein; IL, interleukin; TNF-α, tumor necrosis factor-α, and MCP-1, monocyte chemoattractant protein-1.

**Figure 6 nutrients-11-02919-f006:**
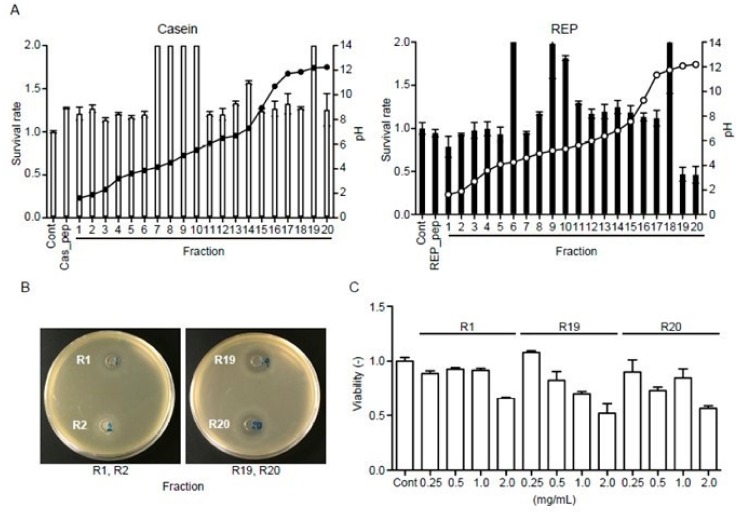
Antimicrobial activity assays for artificially digested casein and REP against *Escherichia coli*. (**A**) Antimicrobial activity of each fraction separated by electrofocusing. The survival rate of *E. coli* is expressed as the A650 ratio of peptide-containing culture medium to medium without peptides. Survival rate (bar) and pH values (open circle) of each fraction from casein or REP hydrolysate samples are shown. Peptide fractions 1, 19, and 20 of hydrolyzed REP exhibited antimicrobial activity against *E. coli*. (**B**) Representative findings of disc diffusion assay plates showing the halos in the bacterial lawn resulting from the antimicrobial activity of the fractions. Mueller-Hinton Agar plates were spread with 0.1 mL from a 10^8^ cfu/mL bacterial suspension. Filter paper discs (6 mm in diameter) were impregnated with 5 µL of undiluted fraction samples and placed on the inoculated plates and incubated first at 4 °C for 2 h and then at 37 °C for 24 h. The diameters of the inhibition zones were measured in millimeters. All tests were performed in triplicate. As shown in A, fractions 1, 19, and 20 (R1, R19, and R20, respectively) exerted antimicrobial activity against *E. coli*, but fraction 2 (R2) did not. (**C**) Antimicrobial activity of 0.25, 0.5, 1.0, and 2.0 mg/mL solutions of fractions 1, 19, and 20 (R1, R19, and R20, respectively) showed a concentration-dependent tendency. Data are expressed as the mean ± standard deviation of three separate experiments. Cont, control (without peptides); Cas pep, total casein peptides, and REP pep, total REP peptides.

**Table 1 nutrients-11-02919-t001:** Final characterization of the four experimental groups (CC, RC, CR, and RR) and the effects of diet on each parameter.

Parameter	Group				*P* Value		
	CC	RC	CR	RR	Juvenile- Period Protein Source	Adulthood Protein Source	Interaction
Body weight (g)	44.6 ± 2.2	39.3 ± 3.7	39.5 ± 3.7	34.5 ± 2.1	<0.001	<0.001	0.829
SBP (mmHg)	117.0 ± 6.5	106.5 ± 10.0	105.1 ± 7.8	104.5 ± 7.7	0.037	0.010	0.062
Total cholesterol (mmol/L)	3.34 ± 0.56	2.73 ± 0.31	2.69 ± 0.22	2.60 ± 0.40	0.008	0.003	0.045
Triglyceride (mmol/L)	0.37 ± 0.13	0.24 ± 0.10	0.19 ± 0.08	0.15 ± 0.07	0.010	<0.001	0.145
Fasting glucose (mmol/L)	10.0 ± 1.4	8.2 ± 1.5	8.7 ± 1.8	6.8 ± 1.6	0.001	0.012	0.986
Fasting insulin (pmol/L)	0.31 ± 0.11	0.20 ± 0.07	0.22 ± 0.11	0.15 ± 0.02	0.006	0.027	0.498
AUC (10^2^ mmol/L·min)	14.1 ± 5.2	10.5 ± 2.4	11.2 ± 2.4	9.9 ± 2.4	0.038	0.121	0.307
HOMA-IR	2.29 ± 0.85	1.21 ± 0.49	1.45 ± 0.88	0.77 ± 0.23	0.001	0.008	0.391
AGEs (µg/mL)	4.05 ± 0.63	3.19 ± 0.85	3.57 ± 0.79	2.65 ± 0.92	0.001	0.055	0.910
Total fat weight (g)	5.8 ± 0.4	4.7 ± 1.0	4.7 ± 1.1	3.2 ± 0.6	<0.001	<0.001	0.428
Mesenteric fat weight (mg)	772.1 ± 193.0	544.7 ± 164.6	504.4 ± 150.9	291.8 ± 63.4	<0.001	<0.001	0.878

The CC and CR groups were fed normal-fat diet (NFD)-C for six weeks, while the RC and RR groups were fed NFD-R. The CC and RC groups were fed high-fat diet (HFD)-C for another 12 weeks, while the CR and RR groups were fed HFD-R. SBP, systolic blood pressure; AUC, area under the curve for glucose (in the oral glucose tolerance tests); HOMA-IR, homeostatic model assessment of insulin resistance; AGEs, advanced glycation end-products. Total fat includes epididymal, right inguinal subcutaneous, mesenteric, and retroperitoneal fat. Data are shown as the mean ± standard deviation.

**Table 2 nutrients-11-02919-t002:** Final characterization of the two groups fed NFD-C or NFD-R in the juvenile period.

Parameter	NFD-C	NFD-R	*P* Value
Body weight (g)	23.3 ± 1.1	22.9 ± 1.0	0.353
Total fat weight (mg)	761.0 ± 146.4	814.5 ± 162.4	0.449
Mesenteric fat weight (mg)	159.6 ± 35.7	158.2 ± 41.7	0.934
Lean body weight (g)	22.6 ± 1.1	22.1 ± 0.9	0.273
Total cholesterol (mmol/L)	3.0 ± 0.2	2.2 ± 0.5	<0.001
Triglyceride (mmol/L)	0.59 ± 0.20	0.42 ± 0.23	0.098
Glucose (mmol/L)	6.8 ± 1.3	6.6 ± 0.8	0.700

Data are shown as the mean ± standard deviation.

**Table 3 nutrients-11-02919-t003:** Characterization of chronic inflammation parameters in the four groups and the effects of diet on each one.

Parameter	Group				*P* Value		
	CC	RC	CR	RR	Juvenile- Period Protein Source	Adulthood Protein Source	Interaction
Serum							
LBP (µg/mL)	10.5 ± 1.3	8.6 ± 1.1	8.6 ± 1.8	7.7 ± 1.4	0.006	0.004	0.270
Leptin (nmol/L)	3.18 ± 0.46	2.19 ± 0.40	2.07 ± 0.57	1.19 ± 0.46	<0.001	<0.001	0.697
IL-1β (pg/mL)	973.2 ± 344.2	625.1 ± 169.0	684.5 ± 229.2	599.1 ± 157.9	0.006	0.043	0.088
IL-6 (pg/mL)	114.1 ± 34.0	88.2 ± 14.7	89.7 ± 12.0	85.9 ± 12.5	0.028	0.046	0.095
TNF-α (pg/mL)	171.6 ± 139.3	73.5 ± 102.3	111.4 ± 139.3	40.5 ± 29.0	0.022	0.197	0.703
MCP-1 (pg/mL)	1041.8 ± 1218.7	274.7 ± 363.0	516.8 ± 628.7	222.5 ± 118.6	0.024	0.208	0.301
Kidney							
IL-1β (pg/g tissue)	827.8 ± 164.1	698.5 ± 125.2	755.7 ± 119.2	705.3 ± 134.5	0.045	0.455	0.368
IL-6 (pg/g tissue)	251.4 ± 53.4	210.8 ± 53.3	216.8 ± 27.7	190.1 ± 24.6	0.016	0.044	0.606
TNF-α (pg/g tissue)	155.4 ± 26.9	131.3 ± 27.3	144.6 ± 26.5	127.7 ± 23.9	0.018	0.392	0.668
MCP-1 (pg/g tissue)	1698.3 ± 297.0	1387.9 ± 288.2	1487.5 ± 198.3	1404.7 ± 255.4	0.023	0.250	0.179
Liver							
IL-1β (pg/g tissue)	567.0 ± 211.7	400.7 ± 95.9	440.6 ± 134.7	363.0 ± 93.8	0.010	0.077	0.331
IL-6 (pg/g tissue)	322.8 ± 147.8	216.0 ± 24.7	249.3 ± 93.8	173.8 ± 44.7	0.003	0.053	0.591
TNF-α (pg/g tissue)	122.1 ± 38.4	88.6 ± 12.4	98.3 ± 35.7	81.1 ± 18.4	0.008	0.091	0.372
MCP-1 (pg/g tissue)	372.9 ± 88.5	284.0 ± 58.2	317.9 ± 91.8	260.6 ± 60.3	0.005	0.113	0.517

LBP, lipopolysaccharide-binding protein; IL, interleukin; TNF-α, tumor necrosis factor-α; MCP-1, monocyte chemoattractant protein-1. Data are shown as the mean ± standard deviation.

**Table 4 nutrients-11-02919-t004:** Kidney and liver parameters of the four experimental groups and the effects of diet on each one.

Parameter	Group				*P* Value		
	CC	RC	CR	RR	Juvenile- Period Protein Source	Adulthood Protein Source	Interaction
Kidney weight (mg)	411.2 ± 23.0	382.6 ± 12.1	413.1 ± 22.0	375.0 ± 16.8	<0.001	0.634	0.429
UAE (µg/day)	239.2 ± 39.4	182.7 ± 38.9	155.4 ± 57.1	96.6 ± 44.4	<0.001	<0.001	0.939
CCr (µL/s)	6.6 ± 1.4	5.5 ± 1.5	5.5 ± 1.3	5.0 ± 1.8	0.106	0.116	0.556
Tubular vacuolar formation area (%)	13.6 ± 1.4	6.6 ± 1.1	6.9 ± 1.6	4.8 ± 2.0	<0.001	<0.001	<0.001
Glomerular area (µm^2^)	4113 ± 249	3630 ± 245	3603 ± 246	3381 ± 165	<0.001	<0.001	0.080
Mesangial matrix area (%)	31.3 ± 0.6	28.3 ± 0.4	28.1 ± 0.8	26.4 ± 0.8	<0.001	<0.001	0.011
Liver weight (mg)	1753.2 ± 284.0	1327.6 ± 136.3	1432.8 ± 233.3	1274.6 ± 124.6	<0.001	0.007	0.047
Liver TG (mg/liver)	25.5 ± 3.8	15.5 ± 4.3	16.5 ± 4.5	11.6 ± 3.9	<0.001	<0.001	0.056

UAE, urinary albumin excretion; CCr, creatinine clearance; TG, triglyceride. Data are shown as the mean ± standard deviation.

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
