# Peer review of "Rice Endosperm Protein Administration to Juvenile Mice Regulates Gut Microbiota and Suppresses the Development of High-Fat Diet-Induced Obesity and Related Disorders in Adulthood"

_nutrients, 2019, doi:10.3390/nu11122919_

Round 1

Reviewer 1 Report

Dear Authors, 

I must congratulate for your work. It is a very interesting study but please to proceed for some proof corrections:

Line 32-47: rewrite  the abstract using Nutrients Journal formatting (introduction, material and methods, etc) I found 23 % plagiarism rate, so, please rewrite the article. I attach the report for helping you. it is clear that your study is based on the animal model. Please, highlight the importance of your findings in the obesity/diabetes/renal kidney diseases field, in human models. What did you suggest with your findings? please specify what kind of rice you used(red/brown/wild)? there are some differences between the rice species regarding the endosperm composition. there are some articles who describe very nice the microbiota patterns in humans with obesity and diabetes, microbiota linked with nutritional patterns too. Please find these articles and correlate with your findings because it is very important finally to know how rice could improve/not the microbiota map.

Author Response

Response to Reviewer 1 Comments

Point 1: Line 32-47: Rewrite the abstract using Nutrients Journal formatting (introduction, material and methods, etc).

Response 1: We have rewritten the Abstract (page 1, lines 35–49) in accordance with the “Instructions for Authors” of Nutrients. We have also added a description of our main aim and emphasized that this study was based on an animal (mouse) model.

Point 2: I found 23 % plagiarism rate, so, please rewrite the article. I attach the report for helping you.

Response 2: With reference to the report you attached, we have rewritten the relevant sections of the manuscript with the assistance of a professional English editing service. The changes in the revised manuscript are shown using the “Track Changes” function.

Point 3: It is clear that your study is based on the animal model. Please, highlight the importance of your findings in the obesity/diabetes/renal kidney diseases field, in human models. What did you suggest with your findings?

Response 3: We agree that this point requires clarification, and have added the following sentences to the Discussion section (page 16, lines 540–545): REP has been administered safely to human patients undergoing dialysis [53], and it has also shown beneficial effects on metabolic syndrome in adults [54]. Therefore, we suggest that the intake of REP or rice-based diets in childhood may prevent obesity and related disorders in adult humans.

Point 4: Please specify what kind of rice you used (red/brown/wild)? There are some differences between the rice species regarding the endosperm composition.

Response 4: In this study, we used a Japanese normal cultivar of rice, Koshihikari (not red rice, but a polished wild type). We have revised our manuscript to add details of the rice used as a raw material in the Materials and Methods section (page 2, line 91) and the types of proteins contained in REP in the Introduction section (page 2, lines 74–77).

Point 5: There are some articles who describe very nice the microbiota patterns in humans with obesity and diabetes, microbiota linked with nutritional patterns too. Please find these articles and correlate with your findings because it is very important finally to know how rice could improve/not the microbiota map.

Response 5: Childhood is an important period for the formation of the gut microbiota in adulthood, and this process is thought to be influenced greatly by diet. In this study, we examined the gut microbiota with or without REP intake and found differences in the abundance of Proteobacteria. According to previous reports, the gut microbiome of obese and type 2 diabetic patients has a high abundance of Proteobacteria [44]. Furthermore, the gut microbiota map of non-vegetarians had a higher abundance of Escherichia coli compared with that of vegetarians [44]. Therefore, we suggest that there is a link between the decrease in the abundance of Proteobacteria (Escherichia)and the intake of REP or a rice-based diet. We have added sentences addressing this issue to the Discussion section (page 15, lines 498–499 and 508–511).

Reviewer 2 Report

In this article, authors reported the “Rice endosperm protein intervention during childhood regulates gut microbiota and suppresses the development of high-fat-diet–induced obesity and related disorders in adulthood” as an original article.

The topic ideal is not a new one and should be updated, especially the animal species (mice) should be mentioned in Topic. Furthermore, the solid information of exact mechanism of rice endosperm protein in energy expenditure and lipid regulation in vitro studies should be provided. However, such information is lack and needed to be clarified.

The composition of rice endosperm protein should be provided to explain the possible mechanism of regulation by antibacterial property of REP in mice guts. The gut microbiota alternation in various AA-contained diets could not provide the real actions of REP in physical regulation of gut dysbiosis. Such information should be provided in Discussion section.

Overall, the article should be reorganized. Authors should provide the potential biochemical findings to connect the possible hypothesis for this study. I do not recommend it in current status.

Author Response

Point 1: The topic ideal is not a new one and should be updated, especially the animal species (mice) should be mentioned in Topic.

Response 1: In accordance with your suggestion, we have rewritten the Abstract (page 1, lines 36–48). We have added a description of our main aim and emphasized that this study was based on an animal (mouse) model. In addition, we changed the title of this paper to “Rice endosperm protein administration to mice during childhood regulates gut microbiota and suppresses the development of high-fat diet–induced obesity and related disorders in adulthood.”

Point 2: Furthermore, the solid information of exact mechanism of rice endosperm protein in energy expenditure and lipid regulation in vitro studies should be provided. However, such information is lack and needed to be clarified.

Response 2: In the previous version of the manuscript, we described the mechanism underlying the regulation of lipids by REP in the Discussion section (page 15, lines 526–537). Our results also suggest that REP peptides have the ability to bind to bile acids, as reported for soybean peptides [50], because a large amount of bile acid was excreted in the feces of the mice administered REP. We would like to show the details of these findings, which are currently being verified using in vitro studies, in a future report. In contrast, there was no significant difference in energy consumption between the mice administered casein or REP, as shown in Table S3 and in the Results section. We agree that the relationship between energy consumption and REP peptides is very interesting, but we would like to address this point in the future as we feel it is beyond the scope of the present study.

Point 3: The composition of rice endosperm protein should be provided to explain the possible mechanism of regulation by antibacterial property of REP in mice guts. The gut microbiota alternation in various AA-contained diets could not provide the real actions of REP in physical regulation of gut dysbiosis. Such information should be provided in Discussion section.

Response 3: As you suggested, the composition of REP was provided in Table S2. We examined the effects of the amino acid composition of casein or REP, but there were no significant differences in body, fat, liver, and kidney weights of mice, as shown in Figure 3. Thus, in this study, we suggest that the effects of REP intake on the alteration of the gut microbiota and the prevention of obesity and related disorders were not due to differences in amino acid composition, but were due to the properties of REP digestion-derived peptides. Indeed, some REP hydrolysate fractions showed antibacterial activity (Figure 6). We have rewritten these points in the Discussion section (page 15, lines 520–5265The identification of the detailed structures of the peptides is a future goal for our group.

Point 4: Overall, the article should be reorganized. Authors should provide the potential biochemical findings to connect the possible hypothesis for this study. I do not recommend it in current status.

Response 4: We agree that it would be important to clarify the potential biochemical findings of REP peptides. However, our current study was designed to focus specifically on the effects of REP intake in childhood using a mouse model. Regarding your suggestion, we have added it as a consideration for future studies (page 16, lines 543–545).

Round 2

Reviewer 1 Report

Thank you for all the corrections. The article is now proper for publishing but after the English text editing. 

Author Response

Thank you for your valuable comments. As you suggested, we have again edited our manuscript with the assistance of the professional English editing service. We will submit the certificate issued by the company. The changes in the manuscript are shown by using the "Track Changes" function in Microsoft Word. In particular, the term “childhood” has been changed to “juvenile period”.

Reviewer 2 Report

Authors responsed the concerns, I have no further questions.

Author Response

Thank you for your valuable comments.  
